# Bridging the Gap between Academia and Industry through Students' Contributions to the FIWARE European Open-Source Initiative: A Pilot Study

Javier Conde *, Sonsoles López-Pernas, Alejandro Pozo, Andres Munoz-Arcentales, Gabriel Huecas and Álvaro Alonso

Departamento de Ingeniería de Sistemas Telemáticos, Escuela Técnica Superior de Ingenieros de Telecomunicación, Universidad Politécnica de Madrid, 28040 Madrid, Spain; sonsoles.lopez.pernas@upm.es (S.L.-P.); alejandro.pozo@upm.es (A.P.); joseandres.munoz@upm.es (A.M.-A.); gabriel.huecas@upm.es (G.H.); alvaro.alonso@upm.es (Á.A.)
* Correspondence: javier.conde.diaz@upm.es

**Abstract:** Although many courses in computer science and software engineering require students to work on practical assignments, these are usually toy projects that do not come close to real professional developments. As such, recent graduates often fail to meet industry expectations when they first enter the workforce. In view of the gap between graduates' skills and industry expectations, several institutions have resorted to integrating open-source software development as part of their programs. In this pilot study, we report on the results of the contributions of eleven students to the FIWARE open-source project as part of their final year project. Our findings suggest that both teachers and students have a positive perception towards contributing to the FIWARE open-source initiative and that students increased their knowledge of technologies valued by the industry. We also found that this kind of project requires an additional initial effort for the students as well as for the instructor to monitor their progress. Consequently, it is important that the instructors have previous experience in FIWARE, as many of the students need help during the process.

**Keywords:** STEM education experiences; active and experiential learning; higher education; university/industry/government partnership; university/industry experiences

## 1. Introduction

Recent graduates often fail to meet industry expectations when they first enter the workforce [1–4]. This issue is of special relevance in software engineering and computer science degrees since a large number of skills are required by most entry-level jobs [5] in addition to the theoretical knowledge that is usually taught in undergraduate courses. Graduates' deficiencies are mainly due to the difficulties of reproducing industry-like scenarios in academic settings [6]. Although many courses in computer science and software engineering require students to work on practical assignments, these are usually toy projects that do not come close to real professional developments [7].

According to the Bologna process [8,9], students are required to complete a final year project (FYP) before obtaining a bachelor's or a master's degree. This is a capstone project in which they must put into practice some of the competencies associated with their major. The FYP's workload is measured by European Credit Transfer and Accumulation System (ECTS), being an ECTS equivalent to 25–30 h of work [10]. An FYP accounts for a minimum of 6 ECTS, but it can get as high as 12.5% (around 30 ECTS) of the total number of ECTS in an engineering bachelor's degree, which usually adds up to 240 ECTS; or as high as 25% (ranging from 12 to 30 ECTS) of the total number of ECTS in a master's degree, which can add up to 120 ECTS. Consequently, it requires a significantly greater dedication on the part of students than any other course in the curriculum. Throughout the FYP, students

are supervised by one or more instructors. One of the two instructors may come from the industry, but at least one of them must belong to the university. In the computer science and software engineering contexts, students' FYPs usually consist of some sort of standalone software application developed in an autonomous way using the technologies of their choice. Each student develops a completely different project, so it is a good opportunity to stand out from their classmates. In spite of the hard work that students put into their FYPs, most of the time these are still toy projects whose complexity does not come close to real-life systems and that have no continuation after graduation.

In view of the gap between graduates' skills and industry expectations, several institutions have resorted to integrate open-source software development as part of undergraduate courses [11] in order to give students the opportunity to participate in real projects and gain some experience in developing code as a part of a consolidated team. Open-source projects usually have a roadmap with a list of new features and issues for which they need volunteer developers. The tasks involved are usually of varying difficulty, so there are many opportunities for newcomers to get involved. However, previous studies have pointed out that a considerable amount of time is needed in order to make meaningful contributions to these projects since the learning curve for students can be quite steep [12–14]. The FYP seems like the perfect opportunity for students to finally engage in industry-like developments, and not only in terms of acquiring technical and soft skills but also related to motivation. Previous studies suggest that students felt more engaged in project-based learning [15]. This can be easily achieved through making substantial contributions to open-source projects since they can dedicate a significant amount of time to its development and the wide assortment of open-source projects available makes it possible to put into practice most of the technical skills that they have learned in class or even acquire new ones. Instead of working on their own, students become a part of a software development team and make significant contributions that outgrow the academic setting. Furthermore, the nature of the FYP allows students to develop their project with an internship, earning experience and easing their future incorporation to the industry.

One of the most frequently reported issues in the literature is finding a suitable open-source project that is amenable to students' contributions [11]. In this regard, the FIWARE European initiative [16] could play a central role in giving students the support they need as newcomers to the software industry. FIWARE (https://www.fiware.org/, accessed on 9 May 2021) is a European open platform that aims to provide a set of open-source software components, known as Generic Enablers (GEs), that ease the development of smart applications in several sectors such as education, e-Health, or smart cities. GEs can be easily assembled together and with other third-party components to build these smart applications, each one covering a different technological area such as security, big data, Internet of Things (IoT), and robotics. Therefore, FIWARE provides a rich ecosystem for students to develop their FYP not only by contributing to the GEs but also by using them to implement their own smart applications.

This pilot study reports the experience of incorporating the FIWARE ecosystem in FYPs. The main aim of the study is to contribute to the existing body of knowledge with the lessons learned from this experience, contrasting them with other studies found in the literature, and providing evidence of how open-source projects, and specifically FIWARE, can help bridge the gap between academia and industry. The investigation evaluates the experience of 11 students who developed their FYP within FIWARE from the perspective of the students, their instructors, and a grading committee. This article also aims to help other instructors integrate open-source initiatives into FYPs, proposing a set of learning objectives, showing the potential benefits of contributing, and highlighting the topics which require more follow-up.

The article is structured as follows. Section 2 reviews related work on developing open-source projects in higher education. Section 3 presents the FIWARE open-source initiative. Section 4 describes FIWARE as an enabling environment for developing FYPs. Section 5 covers the methodology used for evaluating students' experience contributing to

FIWARE in their FYPs. In Section 6, the results obtained from this evaluation are presented and discussed. Lastly, Section 7 concludes this article and gives an outline of future work.

## 2. Related Work

Several studies have reported on the specific technical and personal skills that graduates lack at entry-level jobs [1,3,17]. First, regarding technical skills, previous studies have pointed out students' deficiencies concerning software testing, code quality, code maintenance, database knowledge, exposure to software development tools (such as version control), as well as project experience in industrial production environments [1,18]. Moreover, in regard to personal skills, lack of communication, poor reading and writing skills in English, collaboration, and orientation skills are one of the major issues identified in previous studies [1,4].

A promising approach for bridging the gap between academia and industry is the integration of open-source development into the software engineering or computer science curriculum. A recent study confirmed that the use of open-source projects in the classroom provides a concrete experience similar to industry experience [19]. In the systematic mapping study performed by Nascimento et al. [11], a total of 72 articles were reviewed in order to analyze the extent and range of previous research relating to the use of open-source projects in software engineering education. This study defines nine facets that can be used for classifying educational open-source initiatives: software engineering area, research type, learning approach, assessment perspective, assessment type, approach goal, curriculum choice, control level, and project choice. It was found that there were three main combinations of open-source projects use: (1) full control and predefined projects, (2) no control and free choice projects, and (3) inside control with no or almost no project choice for students.

One of the main challenges reported in the existing literature is choosing a suitable open-source project to which students can contribute [20]. One reason for this is that the project should align with the technologies that students already know or, at least, that they would be able to pick up fast [13]. Another reason is that instructors should have prior knowledge of the codebase [21], which becomes infeasible if each student takes part in a different project. The work conducted by Gokhale et al. [22] provides a four-step manual for integrating open-source projects in software engineering courses, addressing aspects such as code complexity, project compilation process, documentation quality, or project modularity. Ellis et al. [23] proposed evaluation criteria for open-source projects in terms of viability of the project, approachability for the students, and suitability to the course objectives. A key finding in the literature is that projects with many small non-critical-path open implementation issues are particularly effective for newcomers [12].

Another significant challenge of open-source development integration into education is how to evaluate students' contributions to the codebase. Delving into contributing to an open-source project, there are two possible approaches. The first one is developing a new feature, improvement or evolution of an open source project with no intention of submitting it to the official repository of the open-source project. In this case, the requirements are less restrictive and are imposed solely by the instructor. The second option is developing with the intention to submit a contribution to the master branch of the open-source project official repository. In this case, the students have to receive the approval of both their instructor and the official committee of the open-source project. In this sense, very few works use criteria to evaluate students' learning based on either outcomes or developed skills [11]. Some studies describe real experiences of applying open-source projects in academic courses. Researches such as the ones conducted by Ellis et al. [13,24] analyze the impact of learning activities with Humanitarian Free and Open Source Software (HFOSS) projects in terms of motivation, learning experience, and career plans. Other studies describe courses where students take different roles (e.g., developers, testers, requirements definition engineers) in real open source project developments [7,19,25]. Researches, such as the one conducted by Braught et al., compare the methodology followed by different universities and colleges in

courses and capstone projects offered to undergraduate students [12]. Some studies focus on the students' perspective [19,25,26], while others collect impressions from instructors and lead project developers [27,28]. Works like [29,30] present the experience in open-source project courses where students collaborate in small groups.

Regarding the role of FIWARE in education, previous works have reported on how the use of this platform can be used for fostering student learning. Studies such as [31,32] have developed tools that focused on improving techniques and resources for teaching. However, these works do not provide guidelines to prepare students for facing real industrial developments. A different perspective has been adopted by [33], in which a four-stage trajectory of how students can implement a fully-working IoT application based on the FIWARE platform is described.

## 3. The FIWARE Open-Source Initiative

FIWARE (https://www.fiware.org/, accessed on 9 May 2021) is a framework of open-source components, called Generic Enablers (GEs), that ease the implementation of smart solutions. FIWARE provides students with multiple options and technologies to develop their FYP. Its main areas are (1) the integration with third-party systems, including interfaces with IoT devices and robotics; (2) management of context data; and (3) visualization, analysis and processing of context information enabling the smart behavior of applications. Table 1 collects the main GEs present in the catalog of FIWARE [34].

**Table 1.** Catalog of principal FIWARE Generic Enablers (GEs) [34].

| GEs | Description |
| --- | --- |
| Context Broker | Management of context information |
| Iot Agents | Integration with IoT systems |
| Keyrock, Wilma, AuthZForce PDP/PAP | Identity management and Access Control |
| Cosmos | Big Data analysis |
| Draco, Cygnus | Scalable data ingestion |
| Kurento | Processing of real-time media stream |
| CKAN extensions | Extensions for CKAN Open Data platform |
| Wirecloud | Data visualization for rich internet applications |

There are many reasons why FIWARE is a suitable platform in which students can develop their FYP. First of all, FIWARE is a stable environment and large enough to address all activities related to any open-source project (e.g., documentation, deployment, testing). As a result, the students are able to apply and/or reinforce the knowledge acquired during their studies and gain new valuable skills. Moreover, the European Commission has established the adoption of FIWARE as one of the policies for Shaping Europe's digital future strategy under the umbrella of its plan for the upcoming years (2019 to 2024) [16,35]. Consequently, nowadays, there are several projects in the industry funded by the European Union that enhance the use of FIWARE's technology. FIWARE has an international community membered by companies, universities, and institutions [36] (e.g., Atos (https://atos.net/, accessed on 9 May 2021), Telefonica (https://www.telefonica.com/, accessed on 9 May 2021), Red Hat (https://www.redhat.com/, accessed on 9 May 2021), Universidad Politécnica de Madrid (https://www.upm.es/, accessed on 9 May 2021)).

Furthermore, FIWARE provides full-guided, easy-to-reproduce tutorials for most of the GEs based on Docker containers. All the GEs are well-documented, including specific information for users, administrators, and developers hosted in Read the Docs (https://readthedocs.org/, accessed on 9 May 2021). These tutorials and documentation are a perfect entry point for all newcomer students to FIWARE.

Students that develop their FYP within FIWARE are involved in all tasks related to the FIWARE ecosystem, not only the implementation of their contribution. They are required to (1) understand legacy code, (2) design software architecture, (3) follow standards (e.g., NGSIv2, NGSI-LD) and FIWARE specific requirements [37], (4) use version control systems,

(5) adhere to code quality standards, (6) test, (7) deploy, and (8) document the code, (9) expand GE tutorials, and (10) participate in FIWARE periodical meetings. Not many of the aforementioned activities are often found in program curricula. Nevertheless, they are necessary and thus highly demanded in the industry. As a result, FIWARE's FYPs allow students to gain valuable experience for their future incorporation in to the workforce.

## 4. Using FIWARE as an Environment for Developing FYPs

This section explains the methodology followed for adopting FIWARE as a platform for conducting FYPs with the aim of reducing the gap between university studies and the industry.

### 4.1. FIWARE's FYPs Facets

We followed the taxonomy based on facets proposed by Nascimento et al. [11] for classifying open-source development approaches in software engineering education. Table 2 summarizes the facets of our methodology for adopting FIWARE for this purpose.

**Table 2.** Classification of the study by facets.

| Facet | FIWARE's Final Year Projects (FYPs) |
| --- | --- |
| Software Engineering Area | General Software Projects |
| Research Type | Case study |
| Learning Approach | Studio-based learning |
| Assessment Perspective | Student and teacher perspective |
| Assessment Type | Reports, software artifacts, survey, and presentations |
| Approach Goal | Learning SE principles/concepts |
| Curriculum Choice | Capstone project |
| Control Level | Inside control |
| Project Choice | Choice list |

The FYPs developed in the context of FIWARE are thematically diverse and quite different among them. Projects focused on the evolution of FIWARE GEs are the most numerous. Alternatively, some projects are focused on designing architectures and building solutions using the tools provided by the FIWARE ecosystem. Students almost always select their FYP from a list that instructors make public. In few cases, the instructor is the one who proposes a topic directly to the student or vice versa.

This study shows the experience of using the FIWARE open-source initiative as a context for developing students' FYPs at Universidad Politécnica de Madrid. These FYPs constitute a compulsory activity for all master's and bachelor's degrees. They take a duration between 300 and 900 h, depending on the number of ECTS they have in the curriculum (12.5 or 30 in this research). The project is an individual and personalized activity for each student with an instructor, where the feedback from the instructor is also continuous and personalized. The study has measured the results from the perspective of the student, the instructor, and an evaluation committee composed of three teachers. The instruments for evaluating include (1) a survey to assess students' perception of the development of the FYP in the scope of FIWARE, (2) a survey to assess instructors' perceptions in the same regard, and (3) grades provided by the evaluation committee and the instructors themselves.

### 4.2. Assigning Fyps

Regarding project choice, students choose their FYP through an open call demand. Instructors publish the project proposals with a brief description on the department website. Most of the projects are not tightly closed in their initial definition. FIWARE offers a wide catalog of technologies (e.g., machine learning, big data, web development, DevOps), allowing students to select the topic they are most interested in. Students are also able to contact an instructor and propose a project on their own. The instructor assesses

the viability of the proposal from an academic point of view and its adequacy to the FIWARE roadmap.

### 4.3. Defining Requirements/Learning Objectives

Lack of students' experience in open-source projects such as FIWARE makes it necessary to design a plan to introduce these types of initiatives in FYPs. The main objective seeks for students to develop the necessary skills to contribute to an open-source project, participating in all the steps (i.e., selecting the feature in which they contribute, following the rules and process of the project, designing, implementing, and testing the proposal, and contributing the proposal to the code base). It is the responsibility of instructors to define a list of learning objectives (LOs) in order to make the FYP experience productive for the students and achieve this goal [27]. Table 3 presents the complete list of learning objectives that students should achieve in their FYP in order to meet the requirements of the industry and how these objectives can be fulfilled within the scope of the FIWARE open-source initiative. Some of the objectives aim to introduce the student to the FIWARE ecosystem (LO1–LO3), others are focused on designing and developing the contribution (LO4–LO9), and others are activities related to documentation (LO10 and LO11). It should be noted that all FIWARE documentation and communication are in English. It is a requirement that both students and instructors master this language.

**Table 3.** FYP's learning objectives.

| LO | General Learning Objective | Learning Objective within an FYP FIWARE |
|---|---|---|
| LO1 | Read documentation | Read the FIWARE general documentation and the specific GEs documentation |
| LO2 | Apply programming knowledge | Implement code in different programming languages (e.g., Scala, Java, Python) and using different technologies (e.g., Apache NiFi, ORMs) |
| LO3 | Understand legacy code | Understand the existing FIWARE's codebase |
| LO4 | Schedule development steps | Draft FIWARE's roadmap |
| LO5 | Design software architecture | Create UML diagrams |
| LO6 | Write good-quality code | Use tools such as Codacy and ESLint |
| LO7 | Test code | Write unit tests using, e.g., Junit and Mocha |
| LO8 | Use version control | Use Git and GitHub |
| LO9 | Deploy services | Deploy their developments using Docker |
| LO10 | Document code | Write GEs documentation in Read the Docs |
| LO11 | Practice technical writing | Write the FYP thesis |

### 4.3.1. Introduction to Fiware (LO1–LO3)

The first steps when carrying out an FYP are focused on introducing the student to FIWARE. Tasks related to reading documentation prevail (LO1). In their first contact with FIWARE, students have at their disposal a set of tutorials about FIWARE's core and its main GEs. Then, when they have acquired some experience within FIWARE's ecosystem, they should deepen the study of the GE with which they will work in their FYP [34]. In the case where the FYP is about the integration of a set of GEs, the students should deepen more in the connection among the components, instead of a concrete GE. This kind of FYP allows the students to work on projects similar to the ones that they can find in the industry [38].

Depending on each case, students might need to learn or reinforce their knowledge of technologies and programming languages (LO2). For example, to work with FIWARE Context Brokers, the students need to know C++, Kotlin, or Java depending on the implementation chosen (Orion, Stellio, or Scorpio, respectively). With the Cosmos GE, students need to know Scala; with Keyrock, JavaScript; with Draco, Java; etc. Knowledge about tools for software management is also required, e.g., Maven, npm, or rpm for dependency management; Docker for deploying services (LO9); Git for managing version control (LO8); etc.

Once students are familiarized with FIWARE and the technologies required, the next task consists in understanding the existing code to which they are going to contribute (LO3). This task represents one of the most different aspects of developing an FYP within the scope of an open-source initiative. The students do not start from an empty development such as in a traditional FYP based on toy projects. Instead, they must understand the existing code and how and where to implement new functionalities. This sort of development is more often found in the industry; therefore, it will be useful for when they finally join the workforce.

### 4.3.2. Designing and Developing the Contribution Proposal (LO4–LO9)

Before they start developing their projects, students have to set the objectives of their contribution and the phases the development will consist of (LO4). It is recommended to follow FIWARE's roadmap in order to increase the probabilities of it becoming an official contribution to the FIWARE's codebase.

Although the instructor analyzes the contribution before proposing an FYP, it is the responsibility of the students to design how to implement it (LO5). The students will use the experience gained in their previous studies to define the software architecture with tools as Unified Modeling Language (UML) diagrams. When developing the contribution, students must follow the GE guidelines, paying special attention to writing good-quality code using the proposed tools for coding formatting (e.g., Lint4j, ESLint, Codacy) (LO6). In parallel, students have to develop specific tests for their contribution and pass all the original tests of the GE to verify that their contribution does not break it (LO7).

FIWARE uses GitHub as its software configuration management tool and Git for managing versions. Students have to deal with large projects and, consequently, every change has to follow the complete contribution process (LO8). They are required to practice working with branches, tags, releases or pull requests, which are not as present in regular student activities but are essential in the industry.

Students also have to perform tasks related to the deployment of services (LO9). GEs can be installed directly over students' own computers or over Docker containers. Students have to test both options, updating them if there is any change and preparing Docker-based scenarios for the GE tutorials.

None of the FYPs involved in this research had a requirement of becoming an official contribution to the FIWARE codebase. However, all the students had to follow the mentioned steps as part of their work. Moreover, if students finally contribute to the FIWARE codebase, this fact would be positively valued by the instructor and the evaluation committee.

### 4.3.3. Documenting the Results (LO10 and LO11)

On the one hand, students have to follow the guides for documenting their contributions to FIWARE (LO10). FIWARE GEs documentation is hosted in Read the Docs (https://readthedocs.org/, accessed on 9 May 2021), a platform that contains the documentation of many open-source projects. They have to follow the requirements defined by FIWARE [37]. All the GEs documentation includes a brief description of the component, deployment and testing guides, information about accessing and using its APIs, and guided tutorials. The students also have to follow the stylesheet of each GE and include its respective badges with general information of the GE and its quality assessment. Students also have to contribute to FIWARE's official documentation. For that purpose, they are required to have technical reading and writing skills, and their instructors have to validate their contributions.

On the other hand, students have to write their FYP thesis (LO11), which is not necessary for the FIWARE contribution but is required for assessing the quality of the FYP. It is a piece of technical writing that includes an introduction, description of previous related work, analysis, methodology followed, results, and conclusions of their project. Students must follow the guidelines of their own degree, and the instructor has to validate the thesis

before it is graded by the evaluation committee. The research of Radermacher et al. suggests that technical reports, in addition to developments, are also crucial as they help students to improve their communication skills, which are in high demand in the industry [1–3].

### 4.4. Monitoring Progress

Defining a learning plan is often not sufficient to ensure good results in the integration of open-source initiatives within FYPs. Furthermore, it is important to assess progress and validate at each step that the objectives have been met. FIWARE is a large project with a rather steep learning curve. Thus, the integration of FIWARE in a capstone project becomes easier than in regular courses where there are some issues such as lack of time or high ratio between the number of students and teachers.

All the monitoring progress processes developed in this study share a set of tools. (1) Regular meetings (virtual or face-to-face) between the instructor and the student take place regularly. In these meetings, the students present the new features they have developed, state their doubts, and discuss the next development steps. The instructor validates the progress and helps the students with their doubts. This kind of communication offers continuous and personalized feedback, and it helps meet all the learning objectives (LO1–LO11). (2) GitHub allows monitoring the code development from the qualitative and quantitative point of view (LO6, LO8 and LO10). Qualitatively, it helps the instructor analyze the commitments made by the student in the repository. Quantitatively, it helps the instructor to estimate the effort hours throughout various statistics such as the number of lines of code added or deleted in the repository and the frequency with which the student made changes in the code. (3) Travis CI is used to test the deployment and new features developed by the student (LO7, and LO9). It consists of a continuous integration software compatible with GitHub and is widely extended in open-source projects. FIWARE traditionally used Travis CI; however, nowadays it is migrating to GitHub actions. (4) Trello is the tool for activity tracking (LO4). It allows creating online Kanban boards through which the student and instructor are able to know and update the status of all activities.

## 5. Evaluation Methodology

This section presents the evaluation methodology followed in the present pilot study to assess the effectiveness of developing FYPs within the FIWARE ecosystem.

### 5.1. Sample

The sample of the pilot study consisted of 11 students from UPM (Universidad Politécnica de Madrid) who developed their FYP within the scope of FIWARE throughout three academic years. The specificity of the context of the study prevented us from recruiting more participants. Out of all of them, 7 (63.6%) were men and 4 (36.4%) were women. Participants were aged between 22 and 29 years, (median = 23 years, interquartile range = 3 years). A total of 6 students (54.5%) were enrolled in the Bachelor's Degree in Telecommunication Technologies and Services Engineering, 3 (27.3%) were enrolled in the Master's Degree in Telecommunications Engineering, and 2 (18.2%) in the Master's Degree in Network and Telematic Services Engineering. Only 5 students (45.5%) had previous experience in the industry and 2 (18.2%) had previous experience participating in open-source projects.

Five instructors were also involved in the study directing the students' work and in the adoption of the FIWARE open-source initiative in FYPs: two professors and three researchers/collaborating teachers from UPM. Four were men and one was a woman. Instructors were aged between 26 and 57 years (median = 31 years, interquartile range = 27 years). In the scope of this study, two of the instructors conducted three FYPs each, another two conducted two FYPs each, and one instructor conducted one FYP. All the instructors had previous experience developing or using FIWARE. One of the instructors had no previous experience directing FYPs, and in four FYPs (36.4%) it was the first time that the instructors supervised an FYP related to FIWARE. All the survey respondents voluntarily consented to participate in this study. Confidentiality and anonymity were assured at all times.

*5.2. Evaluation Instruments*

Four instruments were used in this study: (a) a survey to evaluate students' perceptions on their experience developing an FYP within FIWARE, (b) a survey to evaluate instructors' perceptions on their experience directing an FYP within FIWARE, (c) a grade determined by the instructor, and (d) a grade determined by a committee.

Students answered a survey after the conclusion of their FYP and after they were evaluated. The first section included generic demographic questions, as well as questions about their previous experience in the industry and in open-source projects. Then, the students were asked about their experience on the FYP through multiple-choice questions and a series of statements with which they had to agree or disagree by using a five-point Likert scale. Lastly, the students scored their experience within FIWARE using a five-point Likert scale.

The instructors answered a survey after the conclusion of all the FYPs supervised. The first section included generic demographic questions, as well as questions about their previous experience in FIWARE and conducting FYPs. After that, they answered, for each FYP, a set of multiple-choice questions, open questions, and a series of statements about the FYP result and students' learning experience, which they had to agree or disagree with by using a five-point Likert scale. Lastly, instructors were asked to provide their overall opinion on the FIWARE initiative and its integration in FYPs through a five-point Likert scale.

The evaluation committee consisted of three teachers who belong to the Department of Telematic Systems Engineering, UPM. They graded the students' FYP with a score ranging from 0 to 10 based on the quality of the students' thesis and of a 10–20 min long defense followed by a round of questions.

## 6. Results and Discussion

The present section shows and discusses the results of the evaluation conducted in the present pilot study. First, the three main aspects evaluated through the surveys completed by both the students and the instructors are presented: learning experience, fulfillment of the learning objectives, and overall opinion. Then, the results obtained from the committee's evaluation are reported.

*6.1. Learning Experience*

Results obtained from the students' and instructors' questionnaires regarding learning experience are shown in Figures 1 and 2, respectively. More than half of the students (54.5%) reported they had some previous knowledge that helped them in the FYP (SQ1, median = 4.0, mode = 4). However, none of the students strongly agreed about this affirmation and three (27.3%) disagreed. In turn, the instructors detected that, in nine of the FYPs (81.8%), the students did not have the necessary skills at the beginning of the project (IQ1, median = 4.0, mode = 4), causing, as a consequence, a big initial effort from the students' side. These findings support those of previous works, in which lack of knowledge of specific technologies and tools was identified as one of the main barriers for software developers in their first experiences in the industry [1,6,18]. The steep learning curve of open-source projects was also identified as a challenge for integrating such projects in university courses [12–14]. An interesting result is that regardless of their previous knowledge, all the students considered they had acquired new useful concepts and skills during the FYP (SQ2, median = 5.0, mode = 5).

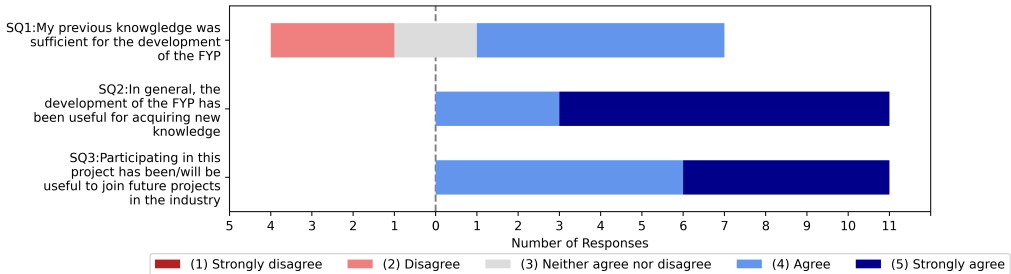

**Figure 1.** Results of student's learning experience general questions.

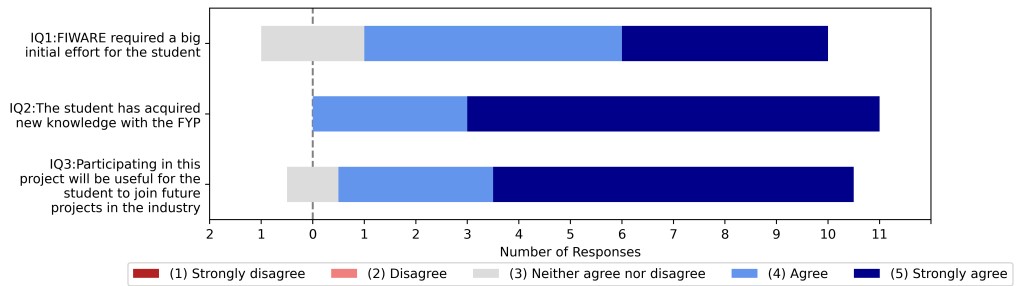

**Figure 2.** Results of instructors's learning experience general questions.

A total of eight students (72.7%) rated this item of the questionnaire with the highest score. Moreover, all the students considered that the FYP would help them in future projects of the industry (SQ3, median = 4.0, mode = 4). These results are aligned with prior investigations, such as the one conducted by Nascimento et al. [7], which found that 90% of the students who participated in an open-source project-based learning course had a similar experience to what they can find in the industry. In addition, in the study by Ellis et al. [24], the students somewhat agreed (mean score of 3.6 out of 5) that they improved their understanding of how to behave like a computing professional through their participation in a Humanitarian Free and Open Source Software project. The instructors also considered that the students acquired new useful knowledge (IQ2, median = 5.0, mode = 5). Specifically, in eight FYPs (72.7%), the instructors strongly agreed with this statement. Moreover, in almost all FYPs (90.9%) the instructors believed the FYP would be useful for their future in the industry (IQ3, median = 5.0, mode = 5). These findings match those of other studies, such as the one conducted by Pinto et al. [28], who interviewed seven professors and concluded that open-source projects are an effective way to introduce new technologies to the student and a good experience for them to include in their resumés. This idea is also consistent with the employers' perspective, who affirmed they are interested in prior project work when they are hiring new graduates [5].

## 6.2. Fulfillment of Learning Objectives

Figures 3 and 4 show students' and instructors' perceptions about the fulfillment of the learning objectives of the FYP, respectively. The literature presents version control (LO8) and testing (LO7) as typical barriers among newcomers [1,3,6,39]. In our research, more than half of the students (63.6%) stated they have learned to use version control systems (SQ4, median = 4.0, mode = 4). However, from the perspective of the instructors, only in three FYPs (27.3%) the students learned it properly (IQ4, median = 3.0, mode = 3). Regarding testing and debugging, more than half of the students (63.6%) answered they have learned to test and/or debug their code (SQ6, median = 4.0, mode = 4). Similar results were obtained from instructors who answered that six of the students (54.5%) acquired testing skills (IQ6, median = 4.0, mode = 3). In four cases (36.4%), the answer from the instructors was neutral, and in just one case (9.1%), the experience was bad. These results match previous studies [19,26] that show the need to deepen the teaching of these topics, especially version control, and they also support the research conducted by Heckam et al.,

which found that when learning testing, the more similar the environment is to a real-world project, the better the results [40].

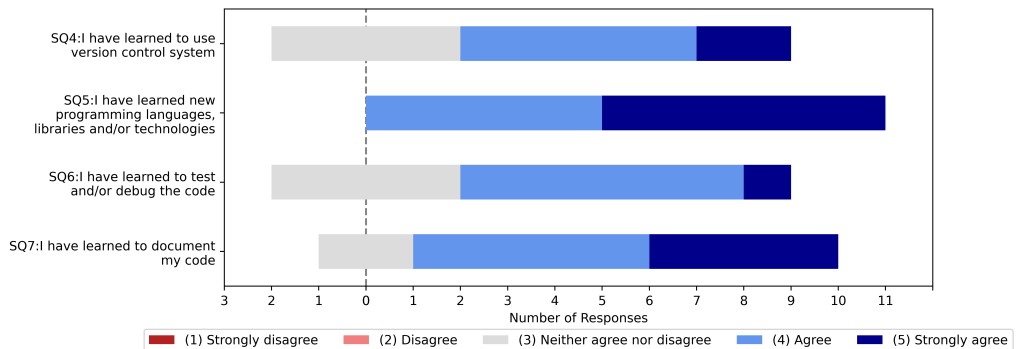

**Figure 3.** Results of learning objectives accomplishment from the students' perspective.

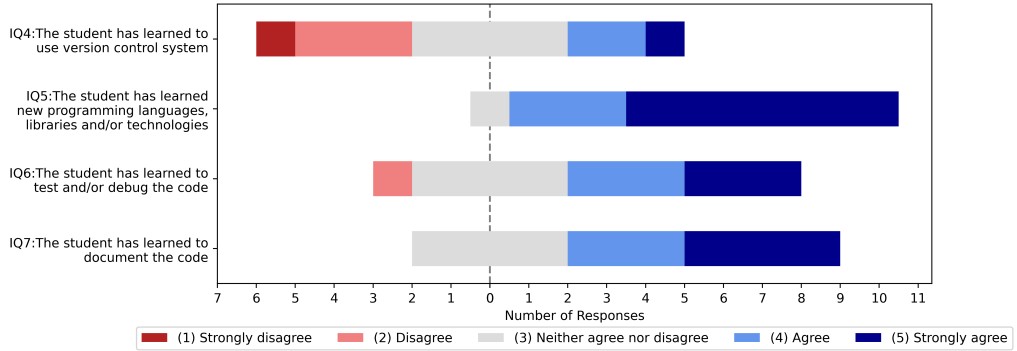

**Figure 4.** Results of learning objectives accomplishment from the instructors' perspective.

All the students learned new programming languages, libraries, and/or technologies (LO2), and more than half (54.5%) rated this statement with the maximum allowed score (SQ5, median = 5.0, mode = 5). From the perspective of instructors, the results were satisfactory too. Just in one case this question was evaluated as neutral, whereas in the rest, the results were positive, with seven answers (63.6%) with the maximum rate (IQ5, median = 5.0, mode = 5). A total of nine students (81.8%) affirmed they learned to document code during the FYP (LO10; SQ7, median = 4.0, mode = 4). The opinion of instructors was less positive (IQ7, median = 4.0, mode = 3, 5). They agreed and strongly agreed that seven students (63.6%) learned this skill, and in the rest of the cases (36.4%), they neither agreed nor disagreed. Other studies revealed similar results, such as the one conducted by Marmorstein [29], where all the students answered that they learned software development tools, and about 90% of them acquired software designing and implementing skills. Another study that matches these results is the one conducted by Nascimento et al. [19], who concluded that with open-source projects, students improve their technical skills, including analyzing, modifying, testing, and documenting code.

Delving into project management, Figures 5 and 6 show the answers of students and instructors about this topic. A total of eight students (72.7%) had problems at the beginning of the development process (LO4; SQ8, median = 4.0, mode = 4), and also in eight cases (72.7%), they required help from the instructor (IQ9, median = 4.0, mode = 5). Regarding time dedication, three students (27.3%) indicated the workload was greater than expected (SQ10, median = 3.0, mode = 2, 3). Similar results were obtained in the instructors' survey, which revealed that in four projects (36.4%), the dedication of the student was greater than initially planned (IQ11, median = 3.0, mode = 3). On the other hand, in eight cases (72.7%), the instructors considered their own workload was greater than the one expected for supervising a regular FYP (IQ10, median = 4.0, mode = 4). Almost all the students (90.9%) were satisfied with the role of their instructor (SQ9, median = 5.0,

mode = 5), in fact, seven of them (63.6%) rated his/her participation with the maximum score on the scale (5). Another interesting result is that instructors' previous knowledge about FIWARE was useful for the students in almost all FYPs (IQ8, median = 5.0, mode = 5). Our investigation produced results that support previous studies that concluded that one of the biggest challenges for students when contributing to open-source projects, or in their first experience in the industry, was the initial impact of contributing to large code bases written by third parties [7,39], caused by the lack of experience in reading legacy code in academic courses [6]. As a consequence, some studies reveal the need of an instructor familiarized with the project to guide the students in the first steps of the project [26,41]. Our investigation confirms these findings. The added difficulty of developing an FYP in the scope of a large open-source project such as FIWARE reveals that in most cases students require more effort to start coding, which could be translated into an increment in the hours devoted and the need of an instructor who is familiarized with the project (i.e., FIWARE).

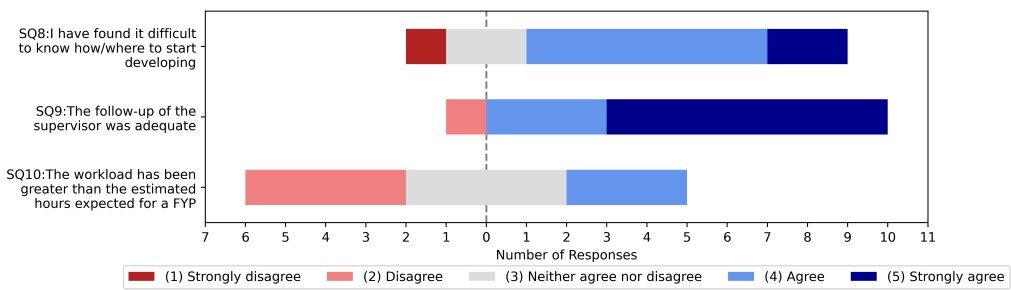

**Figure 5.** Results of project management from the students' perspective.

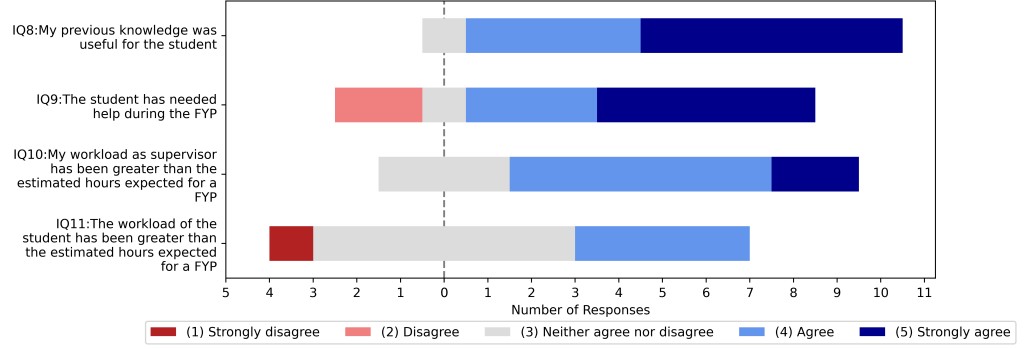

**Figure 6.** Results of project management from the instructor's perspective.

Figure 7 presents the analysis of the most common obstacles from the perspective of the students along the FYP. (1) A total of 54.5% of the students indicated a lack of knowledge about libraries and technologies (LO2, and LO3); (2) 45.5% had problems with the programming languages used in FIWARE (LO6); (3) 45.5% faced problems when writing the thesis (LO11). Other less common obstacles that appeared were (4) lack of time (27.3%); (5) problems with FIWARE's documentation (18.2%) (LO1); and (6) lack of communication with the instructor (9.1%). None of the respondents had problems in tasks related to testing or debugging the code (LO7). From the perspective of the instructors, Figure 8 shows the most common problems observed in the students among which stand out (1) lack of knowledge about libraries and technologies, which affected eight of the students (72.7%) (LO2, and LO3); and (2) lack of time, presented in six FYPs (54.5%). Other obstacles, which affected less than half of the students, were (3) lack of knowledge of programming languages (36.4%) (LO6); (4) missing or incomplete documentation (27.3%) (LO1); and (5) problems structuring and writing the thesis (27.3%) (LO11). Some instructors noted that most students had sufficient knowledge of English to understand the documentation, as well as to design and implement their projects. However, they found that students are not as well prepared to write good-quality and well-structured technical documentation in

English. This is an important point to consider since technical writing is a required skill in most software development environments. Instructors did not detect problems related to testing or debugging the code, or lack of communication with the instructor. From both analyses, it can be concluded that the most typical challenge was the lack of knowledge about libraries and technologies. Again, these obstacles collected in the survey are already present in the literature when describing the barriers of newcomers to the industry [1,6,18].

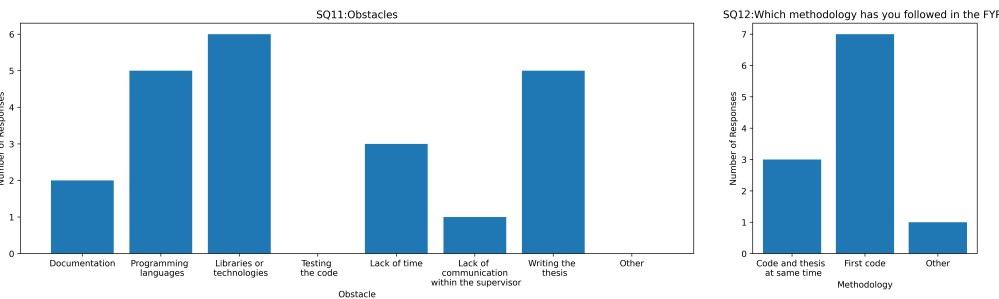

**Figure 7.** Students' obstacles and methodology.

The study also shows in Figure 7 that seven students (63.6%) finished coding before writing their thesis, and only three (27.3%) worked on both activities at the same time. An interesting result is that all the students who pointed out "lack of time" as an obstacle followed the methodology of coding first and writing the thesis last. Conversely, no one who followed the methodology of coding and writing at the same time faced obstacles writing the thesis or experienced lack of time.

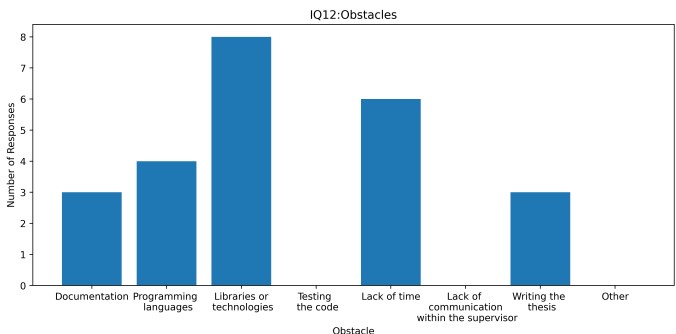

**Figure 8.** Student's obstacles from the instructors' perspective.

### 6.3. Students' and Instructors' Overall Opinion of the Experience

Figure 9 reveals students had positive opinions about using an open-source initiative in their FYP. All of them were satisfied with their work (SQ17, median = 4.0, mode = 4). Specifically, in ten FYPs (90.9%), the students felt very motivated about the fact their contribution had a scope beyond the academic (SQ13, median = 5.0, mode = 5). Most of them (90.9%) would also recommend other students to develop their FYP in an open-source initiative (SQ16, median = 4.0, mode = 4), and nine (81.8%) would repeat the experience if they had to carry out a new capstone project (SQ15, median = 4.0, mode = 4). These positive opinions have also been observed in prior studies [12,13,26], indicating the suitability of open-source projects in general, and FIWARE specifically, as development environments in terms of motivation and learning experience.

On the other hand, the results show that only five students (45.5%) would maintain their code after the FYP (SQ14, median = 3.0, mode = 4). A possible explanation for this result could be that after the completion of their FYP, the students do not have enough time to maintain their development because they are involved in other courses or jobs. This explanation is consistent with the fact that most of the students would repeat the

experience if they had to develop a new FYP and would recommend to other students to develop their FYP in the scope of an open-source project.

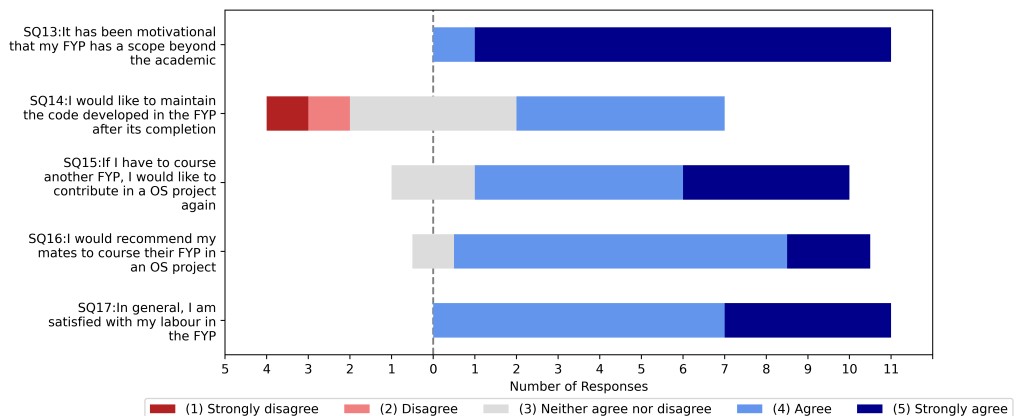

**Figure 9.** Results of student's opinions about using an open-source project in FYPs.

The survey reveals positive results regarding the general opinion from the instructors' perspective (Figure 10). In eight FYPs (72.7%), the instructors were satisfied with their own role (IQ14, median = 5.0, mode = 5). In fact, in six cases (54.5%), they scored the maximum value in the scale (5). Furthermore, in almost all the FYPs (81.8%) they were also satisfied with the results of the project (IQ15, median = 4.0, mode = 5), and in eight cases (72.7%), they were satisfied with the students' work (IQ16, median = 5.0, mode = 5). Another interesting result, in line with the instructors' perception, reveals that eight (72.7%) of the FYPs resulted in a contribution to the codebase of one of the FIWARE's projects. Surprisingly, this result is not aligned with the fact that only five students (45.5%) answered that they would maintain their code in the future (SQ14, median = 3.0, mode = 4). As a consequence of their positive experience, all of the instructors were interested in supervising other FYPs related to FIWARE in the future (IQ13, median = 5.0, mode = 5).

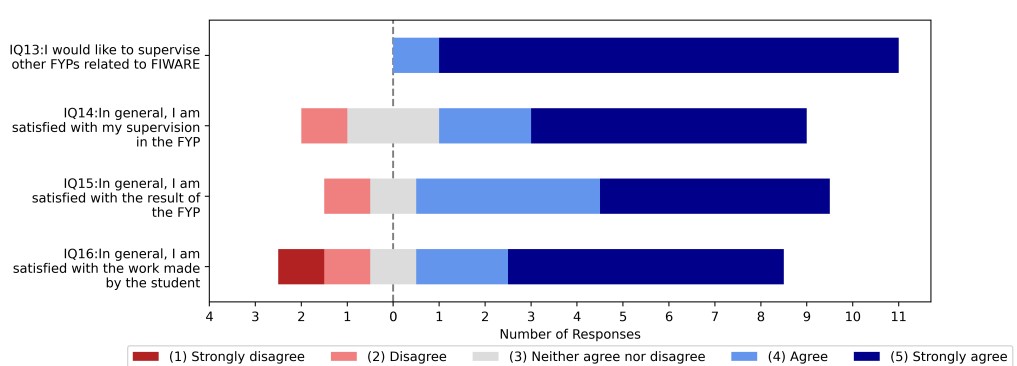

**Figure 10.** Results of instructor's opinions about the FYPs.

### 6.4. Evaluation and Integration of FIWARE in FYPs

Figures 11 and 12 show the results of the survey related to the students' and instructors' opinions about FIWARE and its use in learning activities. In the case of instructors, these questions are independent of the evaluation of the students. Consequently, there is one answer per instructor instead of one answer per FYP supervised, as was the case in the previous subsections.

Almost all the students (90.9%) and all the instructors thought FIWARE GE's are useful tools for industry projects (SQ21, median = 4.0, mode = 4; and IQ19, median = 5.0, mode = 5). Moreover, all of the students were satisfied to have contributed in FIWARE (SQ19, median = 5.0, mode = 5), and eight (72.7%) would like to continue contributing in the future (SQ20, median = 4.0, mode = 4). This result is aligned with previous answers

from the students where they agreed they would repeat the experience (SQ15, median = 4.0, mode = 4). However, it contradicts the result that students are not interested in maintaining their code (SQ14, median = 3.0, mode = 4). A possible explanation could be that it is more attractive for students to develop new functionalities than to maintain code.

Most students and instructors had a positive perception of FIWARE's documentation (SQ18, median = 4.0, mode = 4; IQ18, median = 5.0, mode = 5). The results obtained from our study are aligned with those of Marmorstein [29], who found the documentation of the project was helpful for the students. However, some studies reported lack of documentation and low-quality documentation as one of the common difficulties to adopt open-source projects in educational courses or in projects with newcomers software developers [7,39]. Therefore, although poor documentation is a common barrier hampering the adoption of open-source projects in education, it greatly differs from one project to another.

All the instructors found FIWARE as a good environment for developing an FYP (IQ17, median = 5.0, mode = 5). An interesting result reveals that all the instructors strongly agreed that it is feasible to incorporate FIWARE in regular activities, not only in FYPs (IQ20, median = 5.0, mode = 5). Although the literature explains that capstone projects (e.g., FYPs) are the best option to integrate open-source projects in teaching [27], there are successful experiences of integrating them into other kinds of learning activities [7,13,19,24,25,29,30].

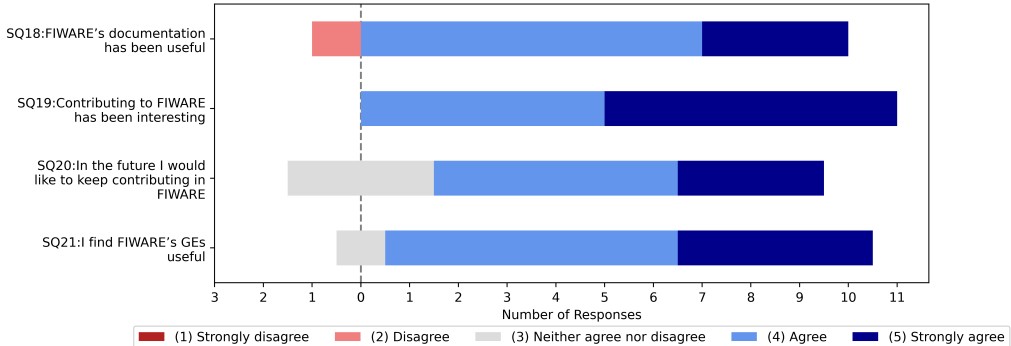

**Figure 11.** Results of student's opinions about contributing to FIWARE in their FYPs.

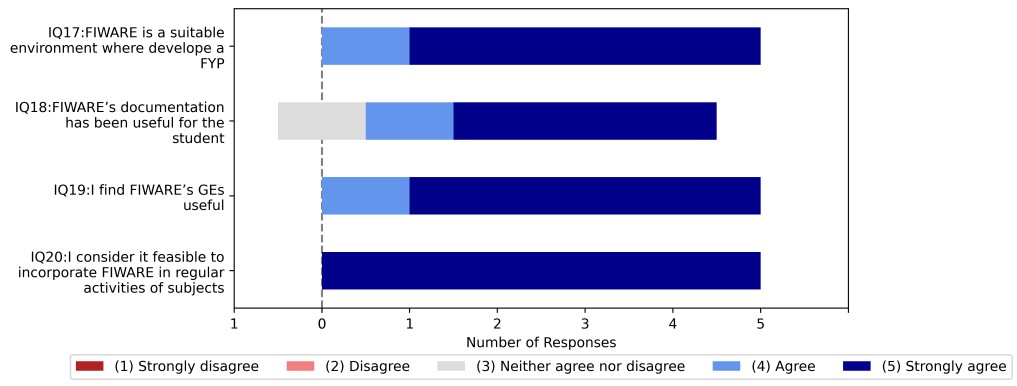

**Figure 12.** Instructors' perception about integrating FIWARE in FYPs.

### 6.5. Students' FYPs Grades

Figure 13 presents the numerical grades provided by the evaluation committees and the instructors of all the FYPs of the study. In both cases, they graded the work from 0 to 10. The committee based their evaluation on the thesis book and a presentation of 15 min by each of the students followed by a round of questions. In turn, instructors graded students by taking into account all the processes of the FYP. In the case of evaluation committees, all the grades ranged between 7.5 and 10 (median = 9.5, mode = 9.5, 10.0) and, in the case of instructors, all the grades ranged between 5 and 10 (median = 9.5, mode = 10.0). The difference of grades between the committee and the instructor (committee grade–

instructor grade) was between $-0.5$ and 2.5 (median $= 0$, mode $= -0.5, 0$). These differences, with the exception of three FYPs, did not exceed more than 0.5 points in absolute value. An interesting result reveals that in these three FYPs, the instructors provided a grade lower than the evaluation committee. A possible explanation for these findings could be that the instructors were better able to determine the weaknesses and strengths of the work, as they supervised the project throughout its entire duration. In contrast, the committee only evaluated the final result through the presentation and the thesis book. In addition, the student required the approbation of their instructor before being evaluated by the committee. As a consequence, the committee provided a smaller range of grades among the students.

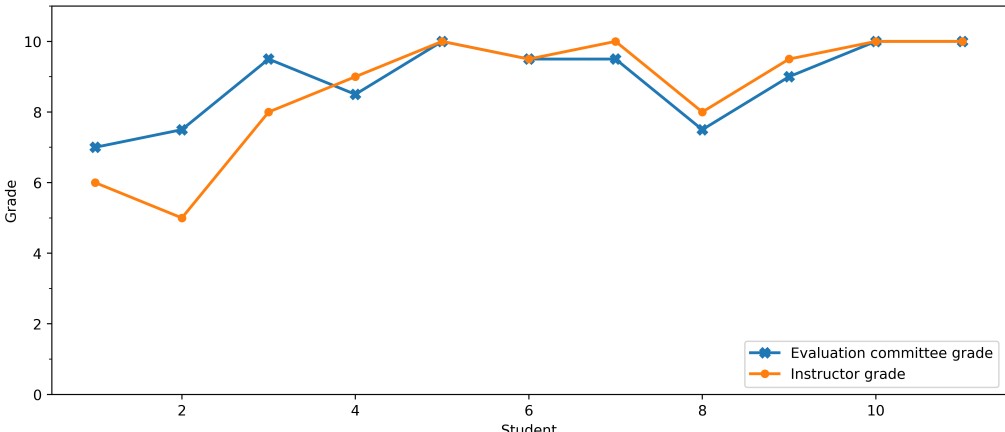

**Figure 13.** FYP grades.

## 7. Conclusions

This article has presented a pilot study at a higher education institution in which students have contributed to an open-source project, namely the FIWARE European initiative, as part of their FYP conducted at the end of their studies. The main goal of this study was to take advantage of the development community in FIWARE to present students with industry-like scenarios so as to prepare them for their imminent incorporation into the workforce. In order to evaluate this experience, we analyzed students' and teachers' perceptions towards the development of FYPs within FIWARE, as well as students' performance. Some important conclusions can be drawn from the study. (1) There is a positive perception towards embracing the FIWARE open-source initiative in FYPs both from students and from teachers. (2) Students increased their knowledge of technologies valued by the industry. (3) Developing an FYP in FIWARE has a significant steep learning curve reflected in the first phases of the project. (4) Students remain interested in contributing to FIWARE after the conclusion of the FYP. From the perspective of the instructors, additional conclusions can be drawn. (5) Instructors believe that developing an FYP with FIWARE prepares the students for their future in the industry. However, (6) this kind of project requires additional initial effort for the students to join FIWARE and for the instructor to monitor their progress. Consequently, (7) it is important that the instructors have previous experience in FIWARE, as (8) many of the students need help during the process.

This pilot study is not without limitations. The most obvious one is the small sample of students. Given that the study was conducted in a single department of a university and that it focuses on a single open-source project, there were no more subjects available for participation in the study. Due to this reduced sample, the study does not include advanced statistical methods that require larger cohorts (e.g., Kendall and Spearman correlations [42]). This is a common problem that appears in other studies in the same field due to the difficulty of finding a representative sample [5,7,18,43]. In contrast, the present article provides important results in terms of the lessons learned from the learning experience and from the comparison with other studies found in the literature. In the future, the instructors

that participated in the study plan to continue offering FYPs within the context of FIWARE to students, which will allow us to extend this sample, analyze the evolution over the years, and gain further insights into the impact of these projects on students' learning. Moreover, the role of students' participation in FIWARE FYPs in paving the pathway towards a smoother integration into the workforce has been assessed solely through students' opinions. Further research is required to determine, in an objective way, whether this is the case, e.g., by carrying out a follow-up of students once they join the labor market. Lastly, another matter in need of further investigation is whether FIWARE has the ability to bridge the gap between academia and industry to a greater extent than other open-source projects.

**Author Contributions:** Conceptualization, J.C. and S.L.-P.; methodology, J.C. and S.L.-P.; software, S.L.-P., A.P., A.M.-A. and G.H.; validation, J.C., S.L.-P., A.P., A.M.-A., G.H. and Á.A.; formal analysis, J.C. and S.L.-P.; investigation, J.C., S.L.-P., A.P., A.M.-A., G.H. and Á.A.; resources, S.L.-P., A.P., A.M.-A. and G.H.; data curation, J.C. and S.L.-P.; writing—original draft preparation, J.C., S.L.-P. and Á.A.; writing—review and editing, J.C., S.L.-P., A.P., A.M.-A., G.H. and Á.A.; visualization, J.C.; supervision, G.H. and Á.A.; project administration, J.C. and S.L.-P.; funding acquisition, G.H. and Á.A. All authors have read and agreed to the published version of the manuscript.

**Funding:** This research was funded by Programa Propio UPM and under H2020-EU.2.1.1.e of the European Commission: FI-NEXT (grant number 732851).

**Informed Consent Statement:** Informed consent was obtained from all subjects involved in the study.

**Conflicts of Interest:** The authors declare no conflict of interest.

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
