# Peer review of "Bridging the Gap between Academia and Industry through Students’ Contributions to the FIWARE European Open-Source Initiative: A Pilot Study"

_electronics, doi:10.3390/electronics10131523_

Round 1

Reviewer 1 Report

The paper presents results of final year students involvement in the development process of OSS project. Merits of such approach to  education are introduced and explained, such as decrease of gap between graduates skills and expectations from industry. The approach is demonstrated on the case of FYP dedicated to improvement of FIWARE library components. Results of social experiment are assessed through the survey. The investigation  showed strong interest to such experience from both side (students and instructors). On the other side, it highlighted lack of skills which are demanded in industrial-scale projects (version control, documenting, testing).

Additionally, 4 questions arise during the reading of the material.

1)  line 30, "... adds up to 240 ECTS". Is it correct? 240 ECTS corresponds to more than 100 working hours per week!  For comparison -  usually employees work 40 hours per week 2)  it is not clear whether an instructor comes from university or from an OSS project development team? 3) the problem of student's English language skills should be addressed - very few students (and unfortunately, university teachers) can speak English well enough to work with documents written in English 4) the question of accepting the FPYs source code from students to an OSS project was not covered. Two situations should be distinguished: A) working stand-alone code; B) a contribution accepted to a master branch. The second one is much harder to achieve, because it requires approval from steering committee of an OSS project or similar organization/department. Especially this question is important in the light of this survey because less than 50% of students plan to maintain their code after FYP.  

The submission provides new data about OSS projects-based collaboration between industry and education. It can be published in Electronics Journal (ISSN 2079-9292) in the present form or with minor changes.

Reviewer 2 Report

Generally, students are sent for industrial training or other words interns in the real work place so as to integrate classroom learning to work place. Similarly, the finals project is aimed at pushing final year students to be creative as well. What difference does it have with this study. Kindly state. However, this is a great study that needs more data and other data analytics models to reach to a more real or practical conclusion.  

  1.  
  2. line 71 to 73 check the grammar and as well make the statement more understandable.
  3. line 81 find out if ''and'' is okay to be there. Move the ''and'' to the 7 section.
  4. check line 98 punctuation
  5. Reference 20 is great. Kindly add more reference to validate/have broader view of the challenges of getting a suitable open source.
  6. What does GE stands for? i see its used frequently  without its full meaning.  line 167; 227; and so on. Kindly define if it is an acronym or ???? also do thesame to other acronym. This helps people of all field to simply understand.
  7. As stated that 11 sample size as limitation. This study could be modified to be called a pilot study or something related or be made s systematic review form.
  8. Find justification regarding the sample size with some reference. 

Round 2

Reviewer 2 Report

Impressive...!